# Lipopolysaccharides (LPSs) as Potent Neurotoxic Glycolipids in Alzheimer’s Disease (AD)

**DOI:** 10.3390/ijms232012671

**Published:** 2022-10-21

**Authors:** Yuhai Zhao, Vivian R. Jaber, Aileen I. Pogue, Nathan M. Sharfman, Christopher Taylor, Walter J. Lukiw

**Affiliations:** 1LSU Neuroscience Center, Louisiana State University Health Science Center, New Orleans, LA 70112, USA; 2Department of Cell Biology and Anatomy, LSU Health Science Center, New Orleans, LA 70112, USA; 3Alchem Biotech Research, Toronto, ON M5S 1A8, Canada; 4Department of Microbiology, Immunology & Parasitology, LSU Health Science Center, New Orleans, LA 70112, USA; 5Department of Ophthalmology, LSU Health Science Center, New Orleans, LA 70112, USA; 6Department Neurology, Louisiana State University Health Science Center, New Orleans, LA 70112, USA

**Keywords:** Alzheimer’s disease (AD), dysbiosis, gastrointestinal tract, lipopolysaccharide (LPS), microbiome, miRNA-30, miRNA-146a, miRNA-155, neurofilament light (NF-L), NF-kB (p50/p65), neuronal atrophy, synaptic disorganization

## Abstract

Lipopolysaccharides (LPSs) are microbiome-derived glycolipids that are among the most potent pro-inflammatory neurotoxins known. In *Homo sapiens*, the major sources of LPSs are gastrointestinal (GI)-tract-resident facultative anaerobic Gram-negative bacilli, including *Bacteroides fragilis* and *Escherichia coli*. LPSs have been abundantly detected in aged human brain by multiple independent research investigators, and an increased abundance of LPSs around and within Alzheimer’s disease (AD)-affected neurons has been found. Microbiome-generated LPSs and other endotoxins cross GI-tract biophysiological barriers into the systemic circulation and across the blood–brain barrier into the brain, a pathological process that increases during aging and in vascular disorders, including ‘leaky gut syndrome’. Further evidence indicates that LPSs up-regulate pro-inflammatory transcription factor complex NF-kB (p50/p65) and subsequently a set of NF-kB-sensitive microRNAs, including miRNA-30b, miRNA-34a, miRNA-146a and miRNA-155. These up-regulated miRNAs in turn down-regulate a family of neurodegeneration-associated messenger RNA (mRNA) targets, including the mRNA encoding the neuron-specific neurofilament light (NF-L) chain protein. While NF-L has been reported to be up-regulated in peripheral biofluids in AD and other progressive and lethal pro-inflammatory neurodegenerative disorders, NF-L is significantly down-regulated within neocortical neurons, and this may account for neuronal atrophy, loss of axonal caliber and alterations in neuronal cell shape, modified synaptic architecture and network deficits in neuronal signaling capacity. This paper reviews and reveals the most current findings on the neurotoxic aspects of LPSs and how these pro-inflammatory glycolipids contribute to the biological mechanism of progressive, age-related and ultimately lethal neurodegenerative disorders. This recently discovered gut-microbiota-derived LPS–NF-kB–miRNA-30b–NF-L pathological signaling network: **(i)** underscores a direct positive pathological link between the LPSs of GI-tract microbes and the inflammatory neuropathology, disordered cytoskeleton, and disrupted synaptic-signaling of the AD brain and stressed human brain cells in primary culture; and **(ii)** is the first example of a microbiome-derived neurotoxic glycolipid having significant detrimental miRNA-mediated actions on the expression of NF-L, an abundant filamentous protein known to be important in the maintenance of neuronal and synaptic homeostasis.

## 1. Introduction

The microbiome of the human gastrointestinal (GI)-tract consists of a dynamic and highly interactive internal prokaryotic ecosystem that possesses a staggering complexity and diversity. Composed of about ~10**^15^** microorganisms, this microbial bionetwork constitutes a significant fraction of the ‘***human metaorganism***’, with considerable commensal and/or symbiotic benefit to the human host. In fact, through extracellular fluid; cerebrospinal fluid; lymphatic and glymphatic dissemination; endocrine, systemic and neurovascular circulation; and vesicular trafficking and/or central and peripheral nervous system (CNS, PNS) passage, this microbiome and its secreted components strongly impact the health, well-being, immunity and vitality, as well as digestive, nutritive and neurological health of the human host [1,2,3,4,5,6,7]. As a complex microbial repository, **(i)** this compartment includes a broad spectrum of microbes, which represent mostly facultative anaerobes or anaerobic bacteria, followed in abundance by archaebacteria, fungi, protozoa, viruses and other microorganisms [1,2,3]; **(ii)** represents the largest source and highest density of microbial species found anywhere in nature [4,5]; and **(iii)** comprises about 1–3% of the human body mass (2 to 6 pounds of microbes in a 200-pound adult) collectively constituting the largest ‘diffuse organ system’ in the human body, and at least as metabolically active as the liver [1,2,3,4,5,6,7]. Relatively recent data from US National Institutes of Health (NIH)-funded interdisciplinary Human Microbiome Project (HMP) initially classified over ~200 thousand diverse, non-redundant prokaryotic genomes in the human GI-tract microbiome comprising about ~5 thousand different GI-tract microbial species, which together encode about ~200 million different protein sequences. Hence, genomic studies of the human GI-tract microbiome indicate that this microbial repository currently represents about ~1 thousand times the genetic complexity of all protein-coding sequences found in the entire human genome [1,2,3,4,5,6,7]. (https://www.hmpdacc.org/hmp/overview/; last accessed on 10 October 2022).

## 2. The Human Gastrointestinal (GI)-Tract Microbiome

The human GI-tract is essentially a ~3.5 cm in diameter and ~7 m long flexible tube that varies in pH and oxygen availability along its entire length. Approximately ~99.5% of all of the resident microbes of the human GI-tract microbiome consist of facultative and/or obligate anaerobic bacteria from just 2 of the 52 major bacterial divisions: *Bacillota* (*Firmicutes*) and *Bacteroides* [1,2,3,4,5]. These form the essential ‘***bacterial-microbial core***’ of the human GI-tract microbiome. The bacterial phylum *Bacillota* (*Firmicutes*) mostly possess a Gram-positive cell wall structure, while *Bacteroides* are an abundant class of obligate anaerobic, Gram-negative pleomorphic-to-rod-shaped bacteria [5,6,7]. As it might be expected, the resident microorganisms of the intestinal epithelium in the deeper and more anaerobic regions of the small intestine are the most enriched in anaerobic and obligate anaerobic microbial species, and these deeper GI-tract regions are heavily populated by the phylum *Bacteriodetes* [1,2,3,4,5,6,7,8,9,10,11]. In these deeper regions of the human GI-tract microbiome, *B. fragilis* can be about ~100-fold more abundant than the phylum *Proteobacteria* and the genus-species *Escherichia coli* [1,2,3,8,9,10,11,12,13,14]. In fact, *Bacteroides* attain extremely high densities in the most anaerobic regions of the human gut at about 10**^10^** to 10**^11^** cells per cubic millimeter of intestinal content and represent the major single source of Gram-negative-bacterium-derived lipopolysaccharides (LPSs) anywhere in the human body ([7,11,13,14]; see below). While GI-tract-derived microorganisms are generally beneficial to human health, when stressed or in pathological states, enterotoxigenic forms of these same microbes have considerable potential to secrete extremely toxic elements, including multiple types of Gram-negative-microbe-derived glycolipids, including LPSs, extremely potent inducers of pro-inflammatory and altered immune-signaling specifically associated with infection and disease.

## 3. The Nature of Lipopolysaccharides (LPSs)

Probably the most well-studied class of neurotoxic species secreted by the human GI-tract microbiome and derived from anaerobic and/or obligate anaerobic Gram-negative bacteria are highly immunogenic, amphipathic glycoconjugates known as LPSs [8,9,10,11,12]. Normally, LPSs have a major natural function to provide structural integrity and a permeability barrier to protect Gram-negative bacteria from microbial toxins and bile salts and from biophysical, biochemical and environmental stress [7,8,9]. Gram-negative bacteria are typically encapsulated by a thin, rigid peptidoglycan meshwork consisting of glycosaminoglycan chains interlinked with short peptides that constitute the bacterial cell wall, and this barrier is surrounded by an outer membrane (OM) embedded with species-specific varieties of LPSs [8,9,10,11,12]. The microbial OM is not a phospholipid bilayer but is instead a highly asymmetric capsule containing phospholipids in the inner leaflet and LPS molecules in the outer leaflet [8,9,10,11,12]. Non-covalently bound and ultimately shed from the OM matrix into surrounding biofluids, large LPS species of approximately 1000 kDa per subunit consist of a hydrophobic lipid-A domain (also known as the LPS endotoxin) attached to a core oligosaccharide (core-OS) and a distal O-antigen (OA), also known as an O-polysaccharide [8,9,10,11,12] (https://pubchem.ncbi.nlm.nih.gov/compound/Lipopolysaccharide; last accessed on 20 September 2022). Interestingly, OA structures, which represent the outermost and most immunologically exposed component of bacterial LPSs, are extremely diverse, and approximately ~200 different serogroups have been identified for *Escherichia coli* alone [3,4,5,11,12]. LPS core-OSs often contain highly immunogenic non-carbohydrate components including amino acids, phosphate groups and/or ethanolamine substituents that are highly variable in composition amongst all Gram-negative bacterial species and even within strains of the same species [5,6,7,9,10,11,12]. Together as a group, **(i)** LPSs are highly varied in their basic structure, organization and immunogenicity; **(ii)** they are classified as pathogen-associated molecular pattern (PAMP) molecules containing microbial-conserved molecular motifs recognized by toll-like receptors (TLRs) and other pattern recognition receptors (PRRs); **(iii)** they activate cells of the innate-immune system, such as macrophages and neutrophils, which synthesize pro-inflammatory factors that include cytokines, chemokines (chemotactic cytokines) and adipokines such as interleukin-1β (IL-1β), tumor necrosis factor (TNF), free radicals and reactive oxygen species (ROS); **(iv)** they act as prototypical endotoxins that promote the efflux of nitric oxides and eicosanoids; **(v)** because of their amphipathic character, they relatively easily pass through GI-tract biophysical barriers into the systemic circulation, especially when these barriers are damaged, diseased or ‘leaky’, such as in ‘***leaky gut syndrome***’ [13,14,15,16,17,18,19,20]; **(vi)** they invariably lead to significant innate-immune and pro-inflammatory responses in the infected host tissue; **(vii)** they strongly induce pro-inflammatory transcription factor signaling, which includes an up-regulation of NF-kB (p50/p65)-DNA binding and the transactivation of pro-inflammatory gene expression, especially in neuroglia and other related brain cell types; **(viii)** they up-regulate the abundance of NF-kB (p50/p65)-sensitive microRNAs, such as miRNA-30b, miRNA-34a, miRNA-146a and miRNA-155; **(ix****)** which is followed by a significant down-regulation of the expression of neuron-specific messenger RNA (mRNA) targets, including those that encode synaptic (SYN), neurofilament (NF) and other structural and cytoarchitectural components of the neuron [9,10,17,18,19,20,21,22,23].

## 4. *Bacteroides fragilis* LPSs and Other Secreted Pro-Inflammatory Neurotoxins

While microbes of the GI-tract microbiome are generally beneficial to global human metabolism, immunity, vitality and health, the enterotoxigenic forms of these same microbes possess significant potential to secrete some of the most neurotoxic and pro-inflammatory biopolymers known [21,22,23,24,25,26,27,28]. Prominent among these noxious bacterial-secreted biopolymers is the LPS class of amphipathic glycolipids/lipoglycans [8,9,10,11,12,13]. These pathogenic neurotoxins were found to significantly disrupt normal gene expression patterns in the CNS and include multiple species of Gram-negative bacteria-derived neurotoxic glycolipids long known to be extremely potent inducers of pro-inflammatory and altered immunological signaling in infection, sepsis, cancer and neurological diseases, including AD [9,10,11,12,13,14,15,16]. Recent research evidence indicates that LPS generation, abundance and secretion is stimulated by a number of factors, including: **(i)** the proliferation of enterotoxigenic LPS-secreting microbes of the GI-tract microbiome and, more specifically, anaerobic Gram-negative bacterial species; **(ii)** metallotoxic environmentally abundant cellular stressors [12,29,30]; and **(iii)** other lifestyle-related inducers, including those commonly found in ‘unhealthy’ Western diets that include high fat and cholesterol (HF-C) consumption and insufficient dietary fiber intake, and other factors [28,31,32,33,34,35,36]. It should be kept in mind that neurotoxic biopolymers originating from the human GI-tract microbiome are solely derived from multiple Gram-negative bacterial species; however, under physiological conditions, very complex combinations of LPSs and other neurotoxins are derived from the entire repertoire of GI-tract resident microorganisms [25,26,27,28,29,30,31,32,33,34,35,36,37,38,39,40,41,42].

The genus *Bacteroidetes*, and in particular Gram-negative anaerobe, non-spore -forming bacillus *Bacteroides fragilis*, which is especially abundant in deeper regions of the GI-tract, is among the most studied and genetically understood of all human GI-tract resident microbes [19,20,21,22,23,24,25,26,27,28,29,31,32,33,37]. *B. fragilis* exhibits a significant amount of intra-species genomic diversity and associated range and variety of potential biochemical functions, and as a prominent *Genus species*, it has significant potential to secrete: **(i)** both a ‘generic’ form of LPS and a unique, exceptionally potent, pro-inflammatory LPS subtype, BF-LPS [30,38]; **(ii)** a zinc-metalloproteinase known as *B. fragilis* toxin (BFT) or *fragilysin;*
**(iii)** truncated LPS molecules known as lipooligosaccharides (*LOSs*) [8,19,20]; and **(iv)** bacterial-derived miRNA-like small non-coding RNAs (sncRNAs) [18,19,20,21]. Bacterial-derived sncRNAs are important microbial regulators that often act to transmit environmental signals when cells encounter suboptimal or stressful growth conditions and whose functions remain incompletely understood [29,30,37,38]. Importantly, LPSs, BF-LPS and/or BFT (*fragilysin*) have significant capabilities to disrupt paracellular and transcellular barriers via the cleavage of intercellular adhesion proteins, resulting in ‘leaky’ and easily breached biophysical barriers [13,14,15,16,17,18]. These barriers: **(i)** become defective, and more penetrable and easily breached with aging and disease; and **(ii)** permit entry of BF-LPS and other LPSs, LOSs, BFT (*fragilysin*), sncRNAs and other microbiome-derived neurotoxins into the systemic circulation, from which they may subsequently access and transit through the blood–brain barrier and gain access to the brain and CNS [14,15,16,17,20,21,22,23].

## 5. LPSs in the Aging and Alzheimer’s Disease (AD) Brain

AD represents a serious, age-related neurodegenerative disorder responsible for a progressive and irreversible behavioral and cognitive decline in our aging population. The neuropathology of AD is based upon the progressive loss of neuron-specific neurofilament (NF) and synaptic proteins, the distortion of neuronal shape and structure, neuronal atrophy, the loss of synapses and neuronal cytoarchitecture, and the appearance of pro-inflammatory amyloid-beta (Aβ) peptides and neurofibrillary tangles in the neocortex [34,38]. GI-tract microbiome dysbiosis or infection by multiple species of toxic bacteria or their secretory products into the brain and CNS was proposed to contribute to the initiation or propagation of the pathogenesis of AD by triggering or accelerating neuro-inflammatory and neurodegenerative responses [38,40,41,42,43]. Less than 6 years ago, several independent reports emerged that detected the presence of GI-tract microbiome-derived LPSs within human brain cells of the CNS both during advanced aging and in the AD brain [27,28,29,31,32,33,37,43,44,45,46,47,48,49]. Interestingly, **(i)** each Gram-negative bacillus, indeed, each GI-tract resident microbe, has the potential to secrete a slightly different LPS and/or neurotoxin assortment with slightly different lipid and oligosaccharide domain structures, abundances, activities, molecular properties and toxicities [17,18,19,20,27]; **(ii)** LPSs have a remarkably high affinity for human neocortical neuronal plasma membranes, and this attraction is significantly enhanced in the presence of the amyloid-beta 42 (Aβ42) peptides that accumulate in the AD-affected brain [15,16,17,18,19,20,31,45] (Figure 1); **(iii)** Gram-negative bacterial molecules associate with AD neuropathology, including amyloid plaques, neurons and oligodendrocytes, in the AD brain [41,42,43,44,45,46]; **(iv)** microbiome-derived *E. coli* LPSs and BF-LPS are associated with the hippocampal CA1 and neocortical regions in AD brain [41,42,43,44]; **(v)** LPS accumulation within neocortical neurons of the AD brain impair the transcriptional output [44,45,46,47,48,49,50]; **(vi)** there is a strong association between LPSs, and neuronal nuclei and the specific LPS-mediated impairment of the expression of the neurofilament light (NF-L) chain gene [50]; and **(vii)** there is a significantly reduced expression of AD-relevant post-synaptic components such as SHANK3 and synapsin-1 (SYN1) in LPS-treated HNG cells in primary culture [8,40,43,49,50]. Importantly, the association between LPSs and the neuronal nuclear envelope of HNG cells in primary culture was also observed in the superior temporal lobe neocortex (Brodmann area A22; Wernicke’s area) and the hippocampal CA1 region of the AD-affected brain [19,32,41,43,51]. In the moderate-to-later stages of AD, LPSs were found to totally encapsulate neuronal nuclei, with the subsequent restriction in the output of genetic information from those specific types of brain cells [16,41,42,43,44,45,46,47,50].

## 6. LPS-Mediated Effects on Neurofilament Light (NF-L) Chain Gene Expression in Alzheimer’s Disease (AD)

The most recent evidence of a neuropathological pathway involving GI-tract-microbiome-derived LPSs and/or BF-LPS, beginning with the LS-mediated up-regulation of pro-inflammatory transcription factor NF-kB (p50/p65) and miRNA-30b signaling, and ending with the specific down-regulation of the neuron-specific cytoskeletal NF-L element is noteworthy [29,39]. This pathway: **(i)** for the first time, underscores a positive pathological link between the LPSs of GI-tract microbes and the disordered cytoskeleton and disrupted synaptic signaling of the AD brain and stressed human neuronal–glial (HNG) cells in primary culture; **(ii)** in part characterizes the up-regulation of NF-kB (p50/p65), a brain-abundant transcription factor known to support inflammatory neuropathology; **(iii)** is representative of a local and progressive pathological signaling system along the gut–brain axis with potential to operate over the life-span of *Homo sapiens*; and **(iv)** it is the first example of a microbiome-derived neurotoxic glycolipid having significant detrimental microRNA-mediated actions on the expression of NF-L, an abundant neuron-specific filamentous protein known to be important for the maintenance of neuronal cell structure and shape, and synaptic homeostasis [27,39,53]. MicroRNA miRNA-30b was previously implicated in targeting inflammasome function [36,39]. Importantly, the sensitivity of a small pro-inflammatory microRNA family consisting of miRNA-30b, miRNA-34a, miRNA-146a, miRNA-155 and others to NF-kB (p50/p65) signaling appears to be in part be due to the presence of one-to-several NF-kB-DNA-binding sites in the immediate promoters of the genes encoding these miRNAs [35,38,39,40,49,50,53,54,55,56]. An improved understanding of LPS molecular genetic interactions along the GI tract–CNS axis involving GI-tract-derived microbial neurotoxins, AD and other related disorders should further support the hypothesis of altered LPS–miRNA–mRNA coupled signaling networks, and these concepts are currently supported by recently described experimental-findings in the scientific literature [29,30,34,35,36,38,39,40,41,42,43,44,45,46,47,48,49,50,53,54,55,56,57]. For example, targeting and/or modulating GI-tract-microbiome-LPS-mediated, miRNA-30b-regulated NF-L pathways and other miRNA-mediated gene expression circuitry could be valuable in the design of future therapeutic strategies that: **(i)** could support and maintain cytoarchitectural components essential for neuronal shape, axonal caliber, inter-neuronal signaling and synaptic plasticity; and **(ii)** may more effectively manage the many neurological diseases in which NF-L gene expression and abundance play a defining and/or determinant role [29,39,55,56,57]. Recent emerging evidence on dietary-based modifications of microbial dysbiosis may also be an attractive means to alter the abundance, speciation and complexity of enterotoxigenic forms of AD-relevant microbes such as *B. fragilis* and their potential for their long-term pathological discharge of highly neurotoxic microbial-derived glycolipid/lipoglycan secretions such as LPS [51,52,58,59,60] (Figure 2).

## 7. Summary

The GI-tract microbiome of *Homo sapiens* is a rich and dynamic source of microbes that possess a staggering degree of diversity and complexity [3,4,5,6,7]. Early studies suggested that the identification and characterization of unique microbial signatures in neuropsychiatric diseases could provide new possibilities in targeted anti- or pro-biotic treatments [31,32]. However, HMP and parallel GI-tract microbiome studies involving multiple human populations and animal models of AD provided evidence indicating that due to wide variation in the microbial and genetic composition of individual GI-tract microbiomes even during healthy aging, it would be difficult to associate the abundance, speciation and/or complexity of any single microbial genus, species or classification with any single human disease [1,2,3,4,5,6,7,8,9,10,45,47,55,59,61,62,63,64,65,66,67,68,69]. This is especially relevant to highly complex and heterogeneous neurological syndromes, such as AD, prion disease and other age-related neurodegenerative disorders, against their complex background of familial genetics, patient age, gender, drug history, age of onset and duration, inter-current disease and other critical environmental and lifestyle risk factors [61,62,63,70].

While commensal GI-tract microbes are beneficial and essential to human health, the enterotoxigenic forms of these same microorganisms have considerable potential to secrete highly neurotoxic biopolymers, including multiple varieties of Gram-negative bacterial-derived glycolipids such as LPSs, which are extremely potent inducers of pro-inflammatory and altered innate-immune and immunological signaling in infection, aging and diseases from AD to cancer [36,39,40,41,42,43,44,45,46,47,48,49,50,51,52,53,54,55,56,57,58,59,60,61,62,63,64,65,66,67,68,69,70,71]. One major characterized pathogenic role of LPSs appears to be the stimulation of cytokine-, chemokine- and/or ROS-mediated pathological signaling programs that drive the induction of pro-inflammatory transcription factor NF-kB (p50/p65), which subsequently promotes the transcriptional up-regulation of NF-kB-sensitive microRNAs. These, in turn, target AD- or related-disease-associated mRNAs, which ultimately down-regulate critically pathologically relevant gene expression programs, resulting in the initiation, development and/or propagation of human disease [21,29,34,49,50,56,63,64,65,66,67,68,69,72,73,74,75]. For example, the LPS-mediated, ROS- and NF-kB-regulated up-regulation of microRNAs miRNA-30b, miRNA-146a and miRNA-155 in transgenic murine models of AD and in AD are now known to target and down-regulate the expression of important neuron-specific neurofilament and synaptic elements important in supporting and maintaining homeostasis in brain cells and neural signaling capabilities [51,52,57,58,59,60,71] (Figure 2). An improved understanding of molecular–genetic signaling along the GI-tract-microbiome–CNS/PNS axis in healthy aging and in AD has significant potential for the development of new diagnostic, prognostic and therapeutic strategies and response-to-treatment efficacy monitoring for the more effective clinical management of AD and other types of progressive, age-related and ultimately lethal neurodegenerative disorders in which LPSs and other microbiome-derived neurotoxins appear to be involved.

## Figures and Tables

**Figure 1 ijms-23-12671-f001:**
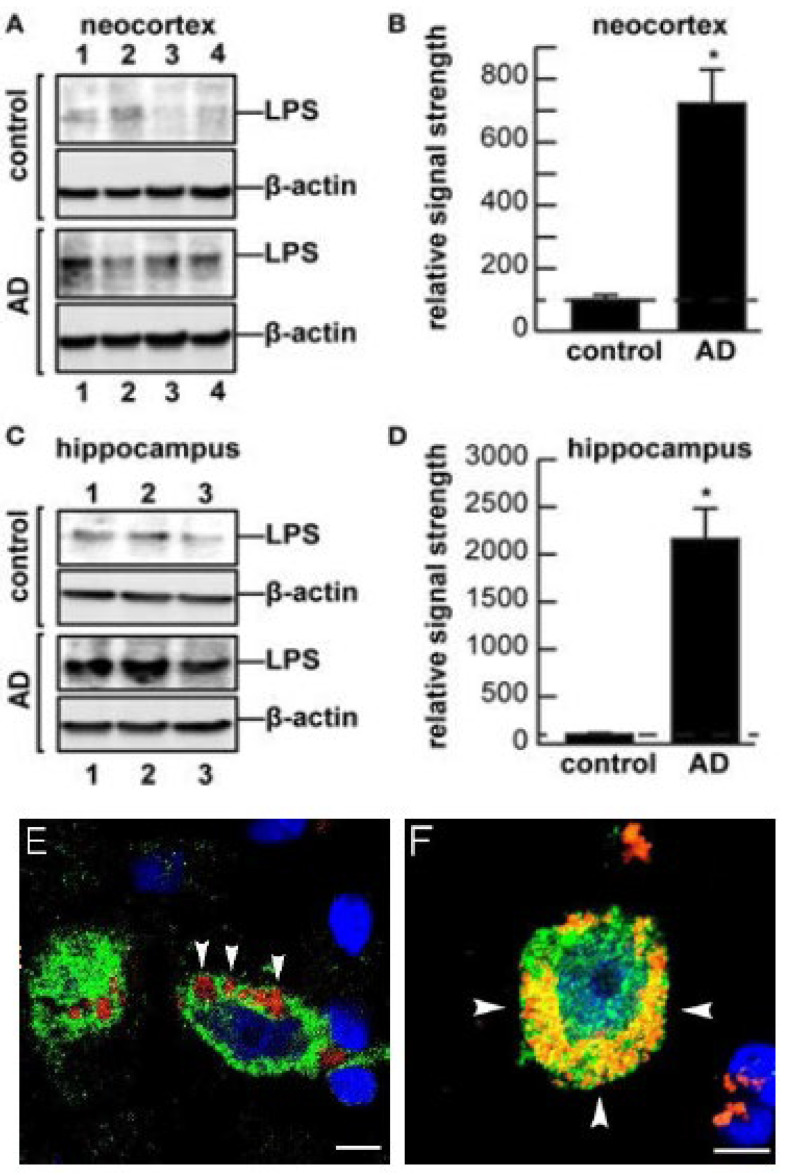
**Affinity of LPS for neuronal nuclei in HNG cells and in AD brain**. (**A**) Western analysis of lipopolysaccharide (LPS; ~37 kDa) signals in human brain temporal lobe neocortex [N = 4 controls and 4 sporadic Alzheimer’s disease (AD) cases; quantified in (**B**)] and in the hippocampus (**C**) [N = 3 controls and N = 3 sporadic AD cases]; quantified in (**D**) compared against β-actin (~42 kDa) abundance in the same samples [using anti-*Escherichia coli* LPSs (cat# ab35654; Abcam, Cambridge, UK) and anti-β-actin (cat# 3700; Cell Signaling, Danvers, MA, USA)]. Methodologies for Western analysis were previously described [41,42,43,45,46,49]. * *p* < 0.05. Densitometric readings of immune-reactive bands were obtained using ImageQuantTL (GE Healthcare, Chicago IL, USA) [12,19,20,41,42,43]; control and AD tissues were age- and gender-matched; there were no significant differences in age (controls, 82.5 ± 8.1 years; AD, 81.3 ± 8.8 years), gender (all female), postmortem interval (PMI; for all tissues, 3.8 h or less), RNA quality or RNA yield between the two groups; in these samples, LPS abundance was found to be on average greater than seven-fold more abundant in AD than the in the control neocortex; LPSs were found to be on average greater than 21-fold more abundant in AD than in the control hippocampus; trends of LPS-nuclear association studies in male brain tissue samples were similar. In (**B**,**D**), a dashed horizontal line at 100 is included for ease of comparison. (**E**) **Immunohistochemical analyses**. Human neuronal–glial (HNG) cells (transplantation grade) in primary co-culture were used to study the dynamics of LPS association with neurons. Blue-stained spherical and oval bodies are DAPI-stained nuclei; image indicates the affinity of LPSs for the neuronal nuclear envelope (white arrowheads); LPSs (red; λmax = 690 nm); neuron-specific β-tubulin III (βTUBIII)-stained (green; λmax = 520 nm) and nuclei (blue; λmax = 470 nm)-stained HNG cells; white arrows indicate punctate and perinuclear clustering of LPSs and LPS affinity for neuron-specific β-tubulin III and the nuclear membrane, as was previously reported [41,42,43,44,45,46,47,48,49]. The incubation of HNG cells with LPSs or BF-LPS gave similar results; in AD, about ~76% of all LPS signals were found to be associated with neuronal cell nuclei; the association of LPS with the major cellular repository for genetic material suggests that the significance of this association may be genetic [41,42,43,44,45,46,47] (adapted from Figure 1B in reference [32]; bottom-right scale bar = 10 µm). (**F**) **Immunohistochemical staining.** Association and envelopment of AD-affected neocortical neuronal nuclei by LPS (red stain; λmax = 690 nm), DAPI (blue nuclear stain; λmax = 470 nm) and NeuN (neuron-specific green stain; λmax = 520 nm); the 3 white arrowheads indicate perinuclear ‘wrapping’ of LPS around neuronal nuclei. This feature has implications for reducing the transcriptional output of neurons [29,50] (microphotograph of sectioned tissue from human superior temporal lobe AD neocortex (Brodmann A22); figure adapted from Figure 3 in reference [52]; bottom-right scale bar = 20 µm).

**Figure 2 ijms-23-12671-f002:**
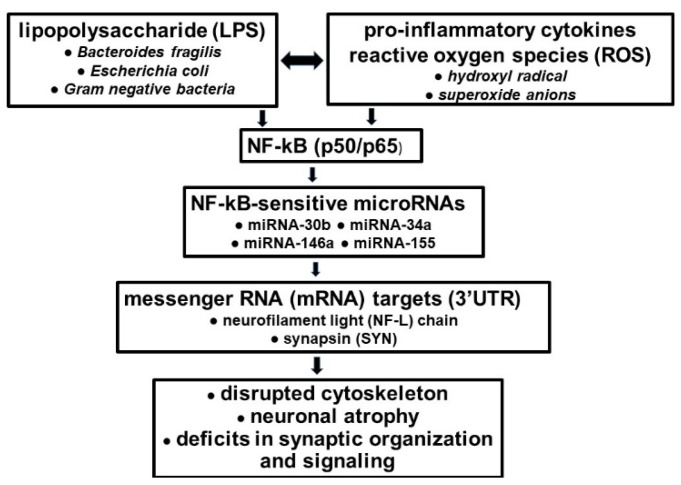
**Highly schematized flow chart of a pathological circuit in AD brain flowing along the gut–brain axis (see manuscript text).** The interactions among GI-tract-microbiome-derived LPSs, pro-inflammatory cytokines and ROS in the generation of the pro-inflammatory transcription factor NF-kB (p50/p65) complex are not completely understood; other miRNAs and mRNAs are expected to be involved in this pathological cascade; importantly, the GI-tract microbiome provides a long-term and life-long source of LPSs; specific components of this pathological pathway provide multiple therapeutic targets for the more effective clinical management of AD.

## Data Availability

All data used in this review are openly available and freely accessible on MedLine (www.ncbi.nlm.nih.gov), where they are listed by the last names of the individual authors.

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
