# Peer review of "Lipopolysaccharides (LPSs) as Potent Neurotoxic Glycolipids in Alzheimer’s Disease (AD)"

_ijms, 2022, doi:10.3390/ijms232012671_

Round 1

Reviewer 1 Report

This review on the role of microbiota-derived LPS in neurological diseases, specifically Alzheimer's disease, and some mechanisms involved, is generally very well written; It is highly interesting and provides relatively new information.

Just a few minor comments:

1. The different sections can be linked better.

Specifically, section 4 begins by summarizing what has been said previously, which may be a bit redundant. I think the first two sentences could be eliminated.

2. On the other hand at different places throughout the text there are additional spaces between words.

3. An issue to comment is that the data shown on the increased presence of LPS in the brain of AD patients belong only to women. It would be very interesting to present similar data for men as well.

Reviewer 2 Report

In the review entitled "Lipopolysaccharide (LPS) as a potent neurotoxic glycolipid in  Alzheimer’s disease (AD)" the authors describe the state of the art about the fundamental role of microbiome and leaky gut syndrome, in the development of AD. The topic is of high interest, and the manuscript overall well written. There are just few  suggestion that I would like to share with the authors. Personally I find quite confusing the way of reporting data as list (i, ii, iii) as in lines 34-40, 117-128 or 200-207. I suggest to the authors to develop a little more those findings in order to write complete sentences that would help the reader in the comprehension. As for chapter 4, I don't see the point of focusing on B. fragilis since the authors have not reported any direct correlation between that specific bacteria type and AD.

Is there any differences in terms of activated pathways depending on LPS type? Are those differences present also in AD? Are there specific bacteria associated with AD development?

Lastly, a couple of figures (one resuming AD connection with the gut, and the other focused on the molecular pathways involved in LPS driven inflammation) might increase the level of this review.

As a minor point there are few typos throughout the text (as misspelling, double spaces... see line 165 double unique; line 187, line 245).

Reviewer 3 Report

The review paper titled "LPS as a potent neurotoxic glycolipid in AD" summarized the research discoveries on neurotoxic aspects of LPS and how it contributes to neurodegenerative disorders, such as AD. 

This paper is well organized and the logic flow is clear. I recommend this paper for publication after 2 minor changes:

1. If possible, please add an illustrative picture or schematic showing the basic idea of this pathological pathway involving LPS

2. In my opinion, the brain-gut-axis should be elaborated a little more. For instance, it was briefly mentioned GI-tract and CNS, PNS in the introduction as well as in the discussion and summary. However, it would be better to elaborate the connection through the brian-gut-axis with more literature support, so that the link between GI tract and neurodegenerative diseases like AD is more solid.

Altogether, the paper provided a very good review of this topic, and the critical comments at the end are fairly thorough. 
